# A Novel Seam Tracking Technique with a Four-Step Method and Experimental Investigation of Robotic Welding Oriented to Complex Welding Seam

**DOI:** 10.3390/s21093067

**Published:** 2021-04-28

**Authors:** Gong Zhang, Yuhang Zhang, Shuaihua Tuo, Zhicheng Hou, Wenlin Yang, Zheng Xu, Yueyu Wu, Hai Yuan, Kyoosik Shin

**Affiliations:** 1Guangzhou Institute of Advanced Technology, Chinese Academy of Sciences, Guangzhou 511458, China; gong.zhang@giat.ac.cn (G.Z.); yh.zhang@giat.ac.cn (Y.Z.); sh.tuo@giat.ac.cn (S.T.); wl.yang@giat.ac.cn (W.Y.); zheng.xu@giat.ac.cn (Z.X.); yy.wu@giat.ac.cn (Y.W.); hai.yuan@giat.ac.cn (H.Y.); 2School of Engineering Science, University of Chinese Academy of Sciences, Beijing 100049, China; 3School of Construction Machinery, Chang’an University, Xi’an 710064, China; 4Department of Robot Engineering, ERICA Campus, Hanyang University, Seoul 426-791, Korea; norwalk87@hanyang.ac.kr

**Keywords:** welding robot, seam tracking, laser sensor, feature point extracting, complex welding seam

## Abstract

The seam tracking operation is essential for extracting welding seam characteristics which can instruct the motion of a welding robot along the welding seam path. The chief tasks for seam tracking would be divided into three partitions. First, starting and ending points detection, then, weld edge detection, followed by joint width measurement, and, lastly, welding path position determination with respect to welding robot co-ordinate frame. A novel seam tracking technique with a four-step method is introduced. A laser sensor is used to scan grooves to obtain profile data, and the data are processed by a filtering algorithm to smooth the noise. The second derivative algorithm is proposed to initially position the feature points, and then linear fitting is performed to achieve precise positioning. The groove data are transformed into the robot’s welding path through sensor pose calibration, which could realize real-time seam tracking. Experimental demonstration was carried out to verify the tracking effect of both straight and curved welding seams. Results show that the average deviations in the *X* direction are about 0.628 mm and 0.736 mm during the initial positioning of feature points. After precise positioning, the average deviations are reduced to 0.387 mm and 0.429 mm. These promising results show that the tracking errors are decreased by up to 38.38% and 41.71%, respectively. Moreover, the average deviations in both *X* and *Z* direction of both straight and curved welding seams are no more than 0.5 mm, after precise positioning. Therefore, the proposed seam tracking method with four steps is feasible and effective, and provides a reference for future seam tracking research.

## 1. Introduction

Mechanical robots have become crucial for modern welding owing to high-volume profitability since manual welding yields low production rates [1]. Robotic welding brings different favorable circumstances, for instance, it has made strides in efficiency, weld quality, adaptability and workspace use, and it diminishes work costs in addition to focused unit cost [2]. 

Be that as it may, most welding robots still work in the working mode of “teach and playback” and their adaptability is not enough when the welding object or other conditions are changed [3]. Since welding as an empirical process is influenced by numerous factors, such as the mistakes of pre-machining, fitting of work pieces, and in-process defects, can result in variation in welding seam. However, welding robots in teach and playback mode have no such capacities and typically weld a weldment with many defects and poor penetration [1].

There are generally three stages in robotic welding: (i) preparation—calibration, robot programming, and weld parameter, work-piece setting, (ii) welding—seam tracking, alternation of weld parameters in real time, (iii) analysis—weld quality inspection [4]. The seam tracking operation is essential for extracting weld seam characteristics which can be fed into the controller of welding robot to instruct the motion of the robot along the welding seam path. Seam tracking technology with laser vision sensing has the advantages of no contact, fast speed, and high precision, which are the keys to realizing welding automation and intelligence [5,6].

In order to fulfill the required welding accuracy for robotic welding, a seam tracking algorithm that enables the robot to plan its path along the actual welding line is necessary. Therefore, many studies have been conducted on automatic seam tracking using sensors such as tactile, touch, probe, vision sensors [7,8], laser sensors [9,10], arc sensors [11,12], electromagnetic sensors [13,14], and ultrasonic sensors [15,16]. The sensors have a very important role in robotic seam tracking; the chief tasks would be weld starting and ending points detection, weld edge detection, joint width measurement.

A basic laser sensor consists of three parts: laser diode, CCD camera, and filter. The laser diode could produce a stripe or dot which would be scanned by the camera. The CCD camera is always fixed at an angle to the laser to capture properly the projection of laser on the work piece [17]. The welding seam tracking system based on laser vision combines laser measurement and computer vision technology. It has the advantages of rich information acquisition, obvious welding seam characteristics, and strong anti-interference ability [18,19], which are suitable for real-time tracking systems. The mathematical model of transforming the laser feature points pixel coordinate to the three-dimensional coordinate of the welding feature points by designing the mechanical structure of the sensor was proposed [20]. 

Chen et al. [21] proposed a feature points positioning method that only needs two profile scans, which can effectively calculate the initial position of the weld. Chang et al. [22] filtered, derived and convolved the weld profile data, and located the feature points by finding the local maxima. Wang et al. [23] established welding seam profile detection and feature points extracting algorithms based on a NURBS-snake and visual attention model, and verified their effectiveness. Mastui et al. [24] introduced an adaptive welding robot system controlled by laser sensor for welding of thin plates with gap variation in single pass.

In a flexible welding process, Ciszak et al. [25] developed a low-cost system for identifying shapes in order to program industrial robots for a welding process in two dimension. The programming of industrial robots was to detect geometric shapes proposed by humans and to approximate them. Based on this, the robot could weld the same profiles on a two-dimensional plane. This is time-consuming as many welding robot applications are programmed by teach and playback, which means that they need to be reprogrammed each time they deal with a new task. Hairol et al. [26] suggested an alternative approach that can automatically recognize and locate the butt-welding position at starting, middle, auxiliary, and end point under three conditions which are (i) straight, (ii) saw tooth, and (iii) curve joint. This was done without any prior knowledge of the shapes involved. As an automatic welding process may experience different disturbances, Li et al. [27] proposed a robust method for identifying this seam based on cross-modal perception so as to precisely identify and automatically track the welding seam.

Wojciechowski et al. [28] proposed the method of automatic robotic assembly of two or more parts placed without fixing instrumentation and positioning on the pallet, which could support a robotic assembly process based on data from optical 3D scanners. The sequence of operations from scanning to place the parts in the installation position by an industrial robot was developed. Suszynski et al. [29] presented the concept of using an industrial robot equipped with a triangulation scanner in the assembly process in order to minimize the number of clamps that could hold the units in a particular position in space based on the proposed multistep processing algorithm.

These efforts have brought about many improvements in the feature points of the target weldment. However, there are certain limitations in the positioning accuracy due the factors such as the change of the welding type (especially oriented to complex welding seam) or the surface defects of the welding.

Due to these circumstances, we here introduce a novel seam tracking technique with a four-step method. First, a laser sensor is used to scan the groove of the weldment to collect profile data; then the data are processed by a filtering algorithm to smooth the noise; next, the second derivative algorithm is proposed to initially locate the feature points based on linear fitting to accurately locate the feature points; finally, according to the results of the sensor pose calibration, the three-dimensional coordinates in the base coordinate system of the welding robot are calculated from the two-dimensional coordinates of the image feature points, and the path planning is completed, with both the line and curve of the Y-shaped groove being targeted as well. The proposed seam tracking technique is tested and verified by way of experimental investigation.

Our proposed seam tracking technique with a four-step method utilizes edge detection and curvature recognition techniques based on laser scan data. The offset of the welding robot’s motion with respect to the welding seam is measured by a laser sensor. By adding a differential point searching method, the feature points of the cross-section of the welding seam are found. Comparing to other seam tracking algorithms, we show the improvement of the required welding accuracy oriented to complex welding seam through theoretical proof, simulation, and experiments. 

This paper is organized as follows: Section 2 presents the seam tracking system composition; Section 3 introduces the seam tracking methodology with four steps; Section 4 shows the results of the experimental investigation based on the proposed seam tracking technique; Section 5 gives the conclusion and perspective.

## 2. Seam Tracking System Composition

The experimental platform composition of the six-axis robot arm for seam tracking system is detailed in Figure 1. As evident in Figure 1, this experimental platform is mainly composed of the motion execution mechanism with six degrees of freedom, laser vision sensor, D/A conversion module, and industrial computer, robotic controller, welding equipment, i.e., welding power supply and wire feeding device, etc.

The execution mechanism is composed of two welding robots, and each of them has six degrees of freedom. The offset of the welding robot’s motion with respect to the welding seam is measured by a laser vision sensor. Through robotic welding experiments, images of molten pool morphology and welding geometry under different welding parameters can be obtained. The main tasks for seam tracking would be weld starting and ending point detection, weld edge detection, joint width measurement, and weld path position determination with regard to welding robot co-ordinate frame.

## 3. Seam Tracking Methodology with Four Steps

In this paper, we introduce a novel seam tracking technique with a four-step method: scanning, filtering, feature points extracting, and path planning. Firstly, the profile information is obtained by scanning the groove with a laser sensor; then, the data are filtered to smooth the noise; next, the feature points are extracted by the combination of the second derivative algorithm and linear fitting; finally, the data of the feature points are converted into the welding seam path of the robot, guiding the welding torch to move and realize the real-time tracking of the welding seam. The flowchart of the proposed four-step method is revealed in Figure 2.

### 3.1. Scanning and Filtering

The purpose of scanning is to obtain the original data of the weldment groove profile, which is the basis for realizing seam tracking [30]. The laser sensor obtains the distance information of the measured object based on the principle of triangulation and then processes the scan data to obtain the profile feature of the measured object. While scanning, the sensor is fixed at the end-effector of the robot and parallel to the welding torch to ensure that the line laser is perpendicular to the measured object [31], covering the groove to the greatest extent, and at the same time, the welding robot is constantly moved to obtain the overall shape of the welding seam.

The combination of limiting filter and Gaussian filter is used to process the groove profile data obtained by scanning. The former is used to remove the pulse interference caused by accidental factors. The latter is used to smooth the data [32]. The data are processed using limiting filtering by comparing the absolute value of the difference between two adjacent sample values and the size of the threshold. Its principle can be expressed as [33]:(1)$$y=\{\begin{array}{cc} y_{n} & \left|y_{n}-y_{n-1}\right|\leq \Delta T\\ y_{n-1} & \left|y_{n}-y_{n-1}\right|>\Delta T \end{array},$$
where *y_n_* and *y_n_*_−1_ are the current and last sampled signal values, respectively, and ∆*T* represents the specified threshold.

Gaussian filtering is a type of linear smoothing filtering method that selects weights according to the shape of the Gaussian function. It is very effective in suppressing the noise that obeys the normal distribution [34], and the Gaussian function has good properties of symmetry, differentiability, and integrability. The function can accurately identify the discontinuous points of the signal, which is very beneficial for the subsequent feature points extracting. The expression of the one-dimensional Gaussian function can be described as [35]:(2)$$f(x)=\frac{1}{\sigma \sqrt{2\pi }}e^{-\frac{{\left(x-\mu \right)}^{2}}{2\sigma^{2}}},$$
where *μ* is the mean value, which determines the position of the function, and *σ* is the standard deviation, which determines the magnitude of the distribution.

### 3.2. Feature Point Extracting

The feature points of the weldment are generally the corner points of the groove section, and its information can reflect the overall situation of the groove profile [36], so feature point extracting is required. This is done according to the cross-sectional characteristics of the weldment groove, combined with the related properties of the function discontinuities listed in Table 1. The groove feature points could be classified as follows: **A**, **B**, **E**, **F**, which are the first type of feature points, and **C**, **D**, which are the second type of feature points, as shown in Figure 3.

Based on the above analysis, the feature points can be located by determining the types of feature points contained in the groove section, and then deriving them to find the extreme points.

#### 3.2.1. Initial Positioning of Feature Points

The preliminary positioning method of the groove feature points is as follows: First, the original data are processed by filtering, and then the first derivative is obtained by the forward difference method and the extreme points are found to determine the first type of feature points, as compared in Figure 4. The abscissa and the ordinate, respectively, represent the *X* and *Z* axes of the sensor coordinate system.

It can be seen from the above figures that the maximum point of the first-order guide falls between the line segment **BC** and **DE**, and fails to accurately correspond to **B** and **E**. This is because the groove of the weldment under actual conditions needs to be machined, and its blunt edge is not a vertical line in an ideal state, but a diagonal line. Therefore, the second type of feature points are transformed into the first type, and the first-order derivative can be continued to differ, and the second-order derivative can be obtained and the point with the highest value can be found to locate all the feature points, as shown in Figure 4. So far, the six characteristic points of the trapezoidal groove have been preliminarily determined, and their location information is listed in Table 2.

#### 3.2.2. Precise Positioning of Feature Points

Due to the defects on the surface of the weldment, as given in Figure 5, the feature points obtained through preliminary positioning are **b** and **c**, while the true feature point should be **a**, which is clearly a deviation. Therefore, on the basis of preliminary positioning, linear fitting is performed on each segment of the groove to accurately locate the feature points.

Suppose any straight-line equation to be fitted is *y* = *ax* + *b*, and the calculation of equation parameters can be written as [37]:(3)$$\left[\begin{array}{c} a\\ b \end{array}\right]={\left[\begin{array}{cc} \sum_{i=1}^{n}x_{i}^{2} & \sum_{i=1}^{n}x_{i}\\ \sum_{i=1}^{n}x_{i} & n \end{array}\right]}^{-1}\cdot \left[\begin{array}{c} \sum_{i=1}^{n}x_{i}y_{i}\\ \sum_{i=1}^{n}y_{i} \end{array}\right],y=\{\begin{array}{cc} y_{n} & \left|y_{n}-y_{n-1}\right|\leq \Delta T\\ y_{n-1} & \left|y_{n}-y_{n-1}\right|>\Delta T \end{array}$$
where *a* is the slope, *b* is the intercept, (*x_i_*, *y_i_*) is the point passing through the straight line, and *n* is the number of points.

The fitting results are shown in Figure 6, and the relevant parameters of the straight line are illustrated in Table 3.

Among them, *SSE* is the sum variance, which calculates the sum of squared errors between the fitting data and the corresponding points of the original data. The smaller the value, the better the fitting affects; *R*-squared is the coefficient of determination, which is used to characterize the quality of the fitting [38]; the closer its value is to 1, the better the fitting affects. It is easy to know that the fitting effect of each straight line is better. The results of precise positioning of the feature points are listed in Table 4. So far, the feature points extracting of the profile for the trapezoidal groove section would be completed.

### 3.3. Path Planning

Because the data measured by the laser sensor are based on their own coordinate system, it is necessary to convert the feature points to the base coordinate system of the welding robot through pose calibration [39].

The relationship between two coordinate systems of the robot is depicted in Figure 7. The sensor calibration is to determine the transformation matrix TSE of {***S***} relative to {***E***}. 

This paper uses the multipoint method for calibration [40]. The main steps are as follows:Select a point *P* on the weldment, make the end of the welding torch this point, and record the position of *P* in the {*B*} coordinate system *^B^**P* = (*x**_B_*, *y**_B_*, *z**_B_*, 1)*^T^*, as shown in Figure 8a.Move the robot so that the laser line of the sensor passes through this point, and record the position of *P* in the {*S*} coordinate system *^S^**P* = (*x**_S_*, 0 *z**_S_*, 1)*^T^*, as shown in Figure 8b.Switch the current tool coordinate system of the robot to {*E*}, record the pose data of the robot at this time, and from the Euler rotation equation, REB can be expressed as [41]:
(4)REB=[cosα−sinα0sinαcosα0001]⋅[cosβ0sinβ010−sinβ0cosβ]⋅[1000cosγ−sinγ0sinγcosγ]=[R11R12R13R21R22R23R31R32R33],
where *α*, *β*, *γ* are the rotation angles of the *X*, *Y*, and *Z* axes of the tool coordinate system {***E***}, respectively.

Then, TEB can be simplified to
(5)TEB=[REBP E0001],
where ***^E^******P*** = (*x**_E_*, *y**_E_*, *z**_E_*)*^T^*, that is, the position of point ***P*** in the tool coordinate system {***E***} after the coordinate system is switched.

According to the transformation relationship of point ***P*** in space:(6)P B=TEB⋅TSE⋅P S,
where the definition of each parameter in the formula is consistent with the above.

Since TSE contains 12 unknowns, at least 3 different fixed points need to be selected to solve the problem. The calibration results in this paper are as follows:(7)TSE=[0.998−0.423−0.59075.098−0.0140.278−0.0266.6930.0020.865−0.814303.1310001],

At this point, the pose calibration of the sensor is completed. For any known points ***^S^******Q*** in its coordinate system, the formula to transform it into the robot base coordinate system can be written as
(8)Q B=TEB⋅TSE⋅Q S,
where ***^B^******Q*** and ***^S^******Q*** are respectively the position of point ***Q*** in the coordinate system {***B***} and the coordinate system {***S***}; TSE is the calibration result of Equation (4); the definition and calculation of TSE follow step 3.

## 4. Experimental Procedures

Experimental demonstration had been carried out at the proposed seam tracking method with four steps to guide the movement of the welding torch under actual testing conditions. Figure 9 reveals the prototype of whole experimental system, which mainly includes ABB IRB 1410 welding robot, IRC5 controller, LS-100CN laser sensor, Ehave CM350 welding power supply, RS-485 communication module, and an industrial computer.

In this paper, two typical weldments with materials of A304 stainless steel are selected as the welding objects, the physical prototypes of two typical welding grooves are illustrated in Figure 10, and the groove parameters of the weldment with straight line and curve are listed in Table 5.

When scanning the welding groove, the laser sensor is set to the trigger mode, and the welding robot is constantly moved to obtain the overall shape characteristics of the welding seam. The process of scanning two typical welding grooves by the laser sensor is represented in Figure 11.

Before the experiment, we mark the starting and ending points of the welding path on the weldment, and then the straight and curved grooves are respectively taught a section of motion trajectory in the model of “teach”, as shown in Figure 10. The red point is the teaching point, which is the position of the end point of the robotic welding torch. Multiple teaching points are connected to form a welding trajectory, and the pose data of the teaching trajectory in the welding torch coordinate system will be recorded simultaneously, which is used as a reference to calculate the experimental deviation.

During the experiment, if the straight groove is taken as an example, let us first move the end-effector of the robot, i.e., the welding torch, along the teaching trajectory. When it reaches reference point **L1**, as shown in Figure 10a, the laser sensor will be turned on to scan the welding groove and collect data. At the same time, the current tool coordinate system of the welding robot will be switched to the end coordinate system, the position and posture data of the end coordinate system are obtained in real time through the API interface of the welding robot, and the sampling period is consistent with that of the laser sensor.

The welding robot continues to move. When the end of the welding torch moves to reference point **L2**, as shown in Figure 10a, the laser sensor will be turned off, the data transmission of the API interface is stopped, the data collection is completed. According to the feature points of the groove, the center point of the welding torch is calculated; according to the position and posture data of the end coordinate system obtained by API interface, the trajectory reference point is calculated. Through the calibration matrix of laser sensor (Formula (7)), the position data of the welding torch center point is transformed into the welding robot end coordinate system, and then through the calibration matrix of welding torch, it is transformed into the welding torch coordinate system.

After the above process, the groove data collected by the laser sensor are transformed into the center point data of the robotic welding torch, and the end coordinate system data collected by the API interface are transformed into the trajectory reference point data. The experimental results of two different welding grooves of straight and curved lines with both initial positioning and precise positioning using the proposed seam tracking method are compared in Figure 12. 

The accuracy of the feature points positioning method is evaluated by comparing the deviation between the calculated welding center point and the actual welding torch end point. Among them, the average deviation *d* (mm) represents the average value of the difference between each welding center point and the end point of the welding torch; the deviation degree *p* (%) indicates the deviation degree of the deviation in this direction relative to the entire groove. The average deviation *d* (mm) and deviation degree *p* (%) can be written as:(9)$$d_{x}=\frac{1}{n}\sum_{i=1}^{n}\left(x_{tcp(i)}-x_{t(i)}\right),\;d_{z}=\frac{1}{n}\sum_{i=1}^{n}\left(x_{tcp(i)}-z_{t(i)}\right),$$
where *d_x_* and *d_z_* are the average deviation in the *X* and *Z* directions, respectively. *x_tcp_*_(*i*)_ and *z_tcp_*_(*i*)_ are the coordinates of the welding center point, *x_t_*_(*i*)_ and *z_t_*_(*i*)_ are the coordinates of the trajectory reference point, respectively. *n* is the number of points.
(10)$$p_{x}=\frac{d_{x}}{l},\;p_{z}=\frac{d_{z}}{h},$$
where *p**_x_* and *p**_z_* are the deviation degrees the in *X* and *Z* directions, respectively. *l* is the total length of the groove, and *h* is the depth of the groove.

The comparative results of different positioning methods for feature points are depicted in Table 6. As can be seen from the figures and table, the average deviations *d**_x_* (mm) of the two different welding seams of both straight line and curve in the *X* direction are relatively large when only initial positioning is carried out. After precise positioning, the average deviations are reduced to 0.387 mm and 0.429 mm, respectively. Experimental procedures show promising results, in that the average deviations display a significant decrease by 38.38% and 41.71%, respectively.

It is worth noting that the average deviations in both *X* and *Z* direction of two different welding seams of both straight line and curve after precise positioning are no more than 0.5 mm; this value is defined by Kovacevic et al. [42] and could fulfill the minimum accuracy requirements of robotic welding. Therefore, it is suggested that the proposed seam tracking method with four steps is feasible and effective, and provides a reference for future seam tracking research.

## 5. Conclusions

A novel seam tracking technique and experimental investigation of robotic welding oriented to complex welding seam are proposed in this study. Conclusions are as follows:A set of seam tracking systems based on laser sensing and visual information extraction is designed, and the method involving scanning, filtering, feature points extracting, and path planning is proposed to realize high-precision seam tracking;The groove information is collected through the laser sensor and the data are filtered, and the corresponding three-dimensional coordinate value in the sensor coordinate system is calculated using the two-dimensional coordinates of the image feature points;The accuracy problem of feature point positioning when the weldment surface has defects is solved. Experimental results show that the average deviations of both straight line and curve of welding feature points after precise positioning is less than 0.5 mm;The experimental errors are mainly caused by the calibration error of the sensor coordinate system and the calculation error of the feature points extracting algorithm. In addition, increasing the resolution of the sensor could further improve the measurement accuracy.

## Figures and Tables

**Figure 1 sensors-21-03067-f001:**
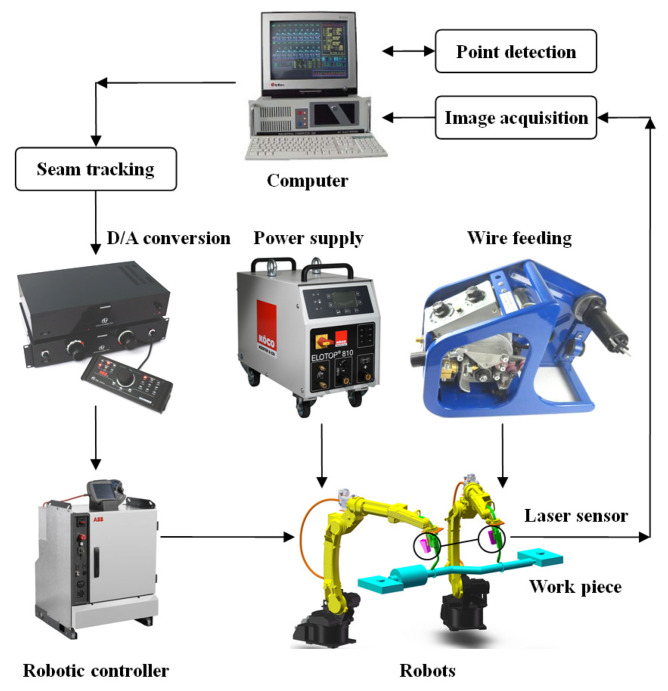
Diagram of seam tracking system.

**Figure 2 sensors-21-03067-f002:**
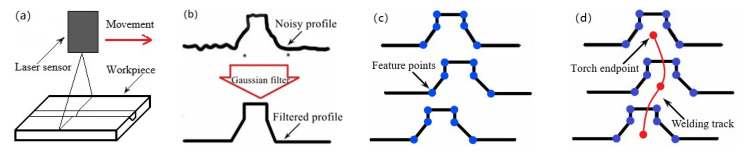
Flowchart of the four-step method for (**a**) scanning; (**b**) filtering; (**c**) feature points extracting; and (**d**) path planning.

**Figure 3 sensors-21-03067-f003:**
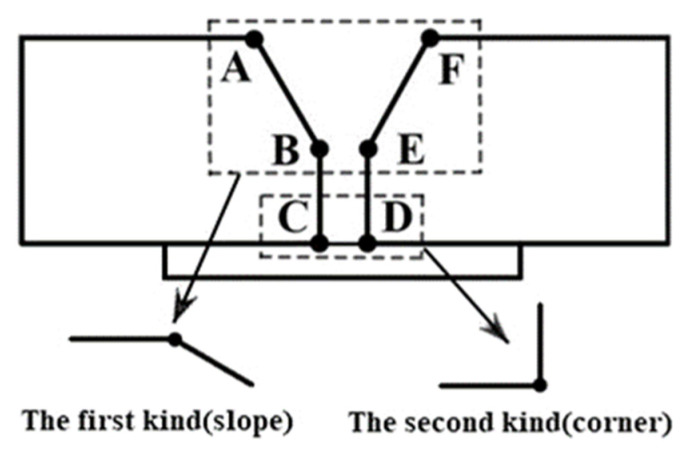
Classification of groove feature points.

**Figure 4 sensors-21-03067-f004:**
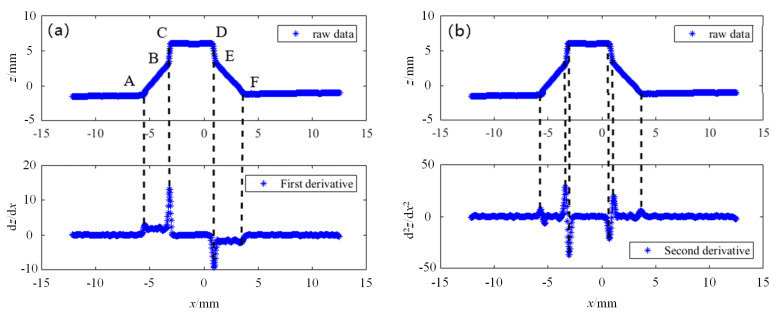
Initial positioning of feature points for (**a**) the first type of feature points; and (**b**) all feature points.

**Figure 5 sensors-21-03067-f005:**
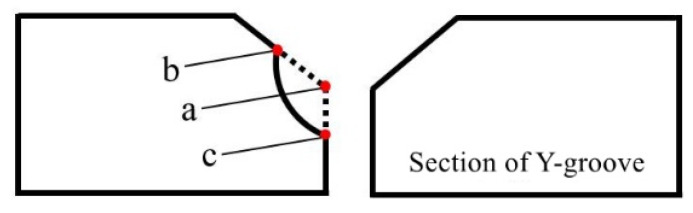
Defects on the surface of the weldment.

**Figure 6 sensors-21-03067-f006:**
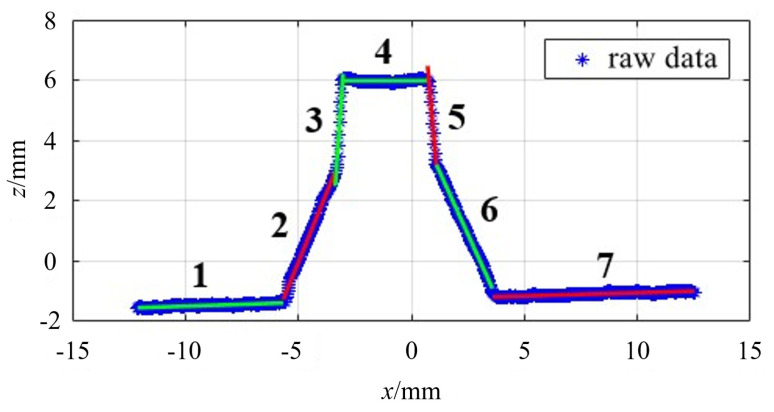
Fitting results.

**Figure 7 sensors-21-03067-f007:**
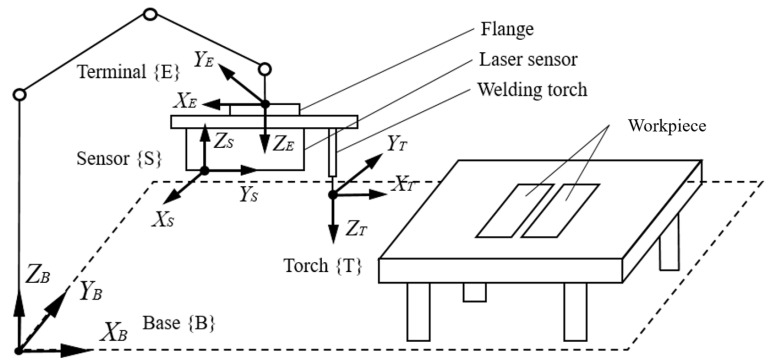
Relationship between two coordinate systems.

**Figure 8 sensors-21-03067-f008:**
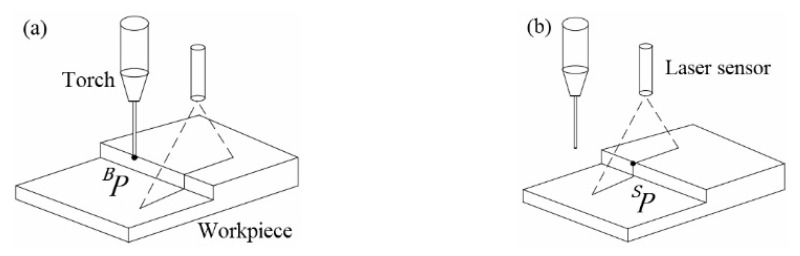
Laser sensor calibration for (**a**) base coordinates; and (**b**) sensor coordinates.

**Figure 9 sensors-21-03067-f009:**
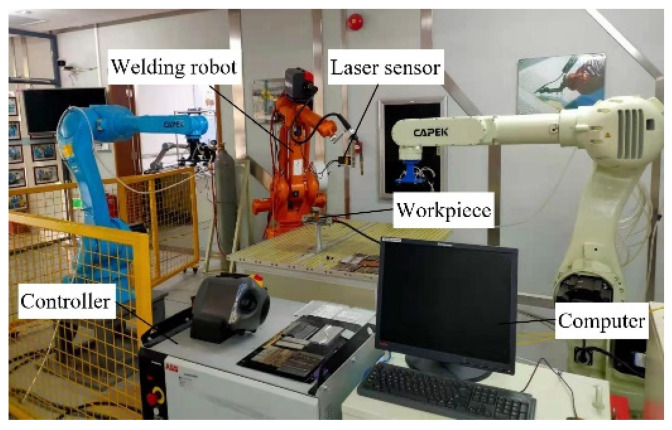
A prototype of the experimental system.

**Figure 10 sensors-21-03067-f010:**
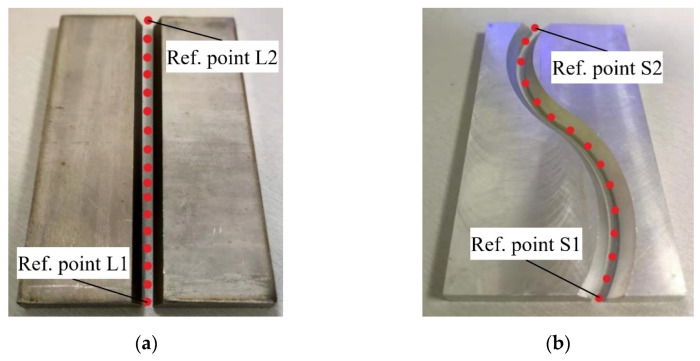
Two typical welding grooves for (**a**) straight line; and (**b**) curve.

**Figure 11 sensors-21-03067-f011:**
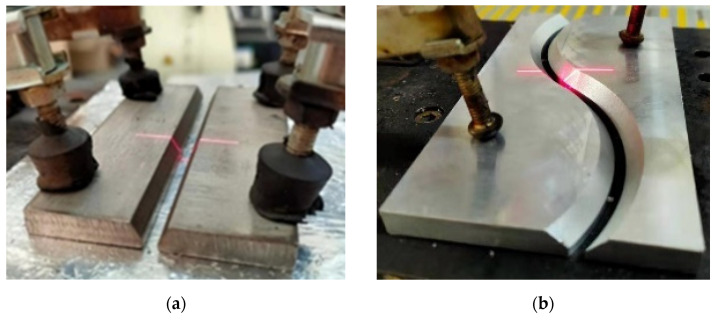
Two typical welding grooves scanned by laser sensor: (**a**) straight line; (**b**) curve.

**Figure 12 sensors-21-03067-f012:**
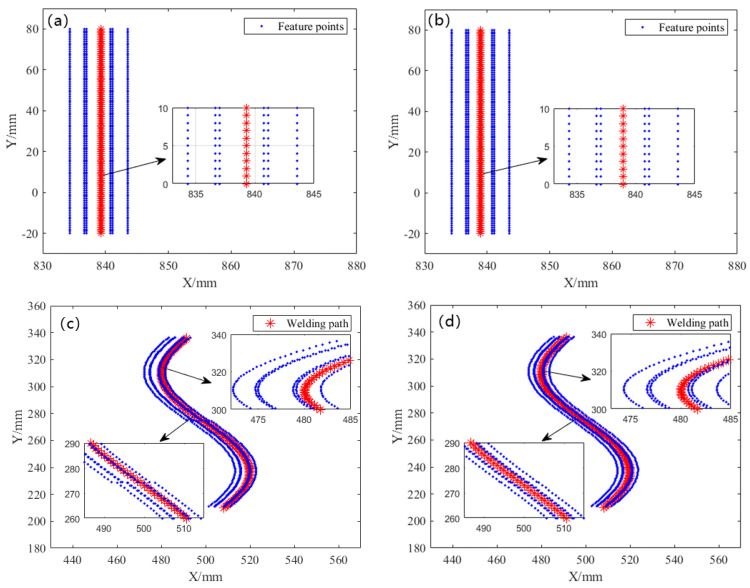
Experimental results of (**a**) straight line with initial positioning; (**b**) straight line with precise positioning; (**c**) curve with initial positioning; and (**d**) curve with precise positioning.

**Table 1 sensors-21-03067-t001:** Properties of discontinuous points of function.

Discontinuous Points Type	Amplitude	First Derivative	Second Derivative
The first	continuity	Step mutation	extremum
The second	continuity	non-existent	/

**Table 2 sensors-21-03067-t002:** Results of initial positioning.

Feature Points	A	B	C	D	E	F
*X*/mm	−5.67	−3.37	−3.02	0.72	1.11	3.59
*Z*/mm	−1.35	2.89	6.03	6.01	3.15	−1.02

**Table 3 sensors-21-03067-t003:** Parameters of fitting straight line.

Fitting Straight Line	1	2	3	4	5	6	7
*SSE*	0.08	0.44	0.39	0.15	0.50	0.15	0.21
*R*-squared	0.85	0.99	0.95	0.87	0.97	0.99	0.81

**Table 4 sensors-21-03067-t004:** Results of precise positioning.

Feature Points	A	B	C	D	E	F
*X*/mm	−5.73	−3.31	−3.04	0.78	1.10	3.76
*Z*/mm	−1.39	3.07	5..98	5..99	3.22	−1.18

**Table 5 sensors-21-03067-t005:** Groove parameters of weldment.

Welding Type	Dimension/mm	Thickness/mm	Slope Angle/°	Blunt Edge/mm
Straight line	100 × 60	8	45	2.5
Curve	130 × 70	10	60	3

**Table 6 sensors-21-03067-t006:** Error analysis results.

Welding Type	Initial Positioning				Precise Positioning			
	*d**_x_*/mm	*d_z_*/mm	*p_x_*/%	*p_z_*/%	*d**_x_*/mm	*d_z_*/mm	*p_x_*/%	*p_z_*/%
Straight line	0.628	0.214	6.688	2.665	0.387	0.230	4.121	2.864
Curve	0.736	0.185	7.838	2.304	0.429	0.251	4.569	3.126

## Data Availability

The data presented in this study are openly available in [A Novel Seam Tracking Technique with A Four-Step Method and Experimental Investigation of Robotic Welding Oriented to Complex Welding Seam—research data] at [https://cloud.huawei.com/home#/collection/v2/all] (accessed on 15 April 2021).

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
