# Peer review of "A Novel Seam Tracking Technique with a Four-Step Method and Experimental Investigation of Robotic Welding Oriented to Complex Welding Seam"

_sensors, 2021, doi:10.3390/s21093067_

Round 1

Reviewer 1 Report

Weld line tracking algorithms are considered in this study, and the subject here is presumably multi-layer welding of MAG. In this welding, the issues are the setting of welding conditions and the perpendicularity to the weld line. I understand the usefulness of this method when the weld line is straight. However, in an S-shaped curve, the torch axis is usually rotated to match the curve in order to pivot the torch perpendicular to the weld line. However, the rotation has not been considered.

Other some questions to ask are as follows

No.1

L45. " in-process defects"

In general, defects in the welding field cannot be corrected by welding operations. What is  "in-process defects"?

No.2

L54

"Seam tracking technology has the advantages of non-contact, fast speed, and high precision,"

Arc sensors are commonly used in arc welding, and these are not called non-contact or high-speed.

No.3

L99  Replace "composed by"  by "composed of", because equipment is not a subject. 

No.4 "the line laser is perpendicular"

  Although it is vertical in Figure 2, it cannot maintain verticality if the welding torch axis rotates. Also, in Fig. 2, it is straight, but in Fig. 11(b), it is S-shaped, so it cannot maintain verticality. It is unclear how the rotation of the torch axis is considered.

No.5

L241 Term "Welding conditions" was described. 

In general, welding conditions refer to welding speed, current, voltage and number of stacks for the groove  geometry, which are not described here.

No.6 

L308  This sentence describes an improvement in accuracy.

In bevel welding, the welding wire is generally bent and not straight. For this reason, the effect of wire bending is more important than the accuracy of the sensor. Considering the accuracy in this study would not be useful in actual welding.

No.7

Welding gun and welding torch are mixed together in this paper. It would be better to use welding torch.

Reviewer 2 Report

The article is weak in terms of comprehensibility and language, correction is recommended.It should be mentioned where in the surface the calibration took place. Is the axis of the tool oriented as it was during testing? Will the test results be reproducible when the groove is in a different surface than during calibration?It seems relatively time consuming to scan and execute the program in two steps. At what speed (in mm / s) is the weld scanned? It should be mentioned whether it is possible to expand the system to enable simultaneous scanning, correcting the position and welding.The novelty of the solution should be more clearly indicated. How this solution differs from other welding path correction solutions available on the market, such as Binzel TH6x.The literature analysis should include if necessary solutions related to  example:programming of industrial robots using the recognition of geometric signs in glexible welding Pprocess oroptical scanner assisted robotic assembly.

Round 2

Reviewer 2 Report

Thanks for the corrections made. In my opinion, however, you should also introduce more literature references at the outset, as I have proposed. Best regards
